# A deep learning approach to predict visual field using optical coherence tomography

**Keunheung Park[1,2], Jinmi Kim[3], Jiwoong Lee[1,2]***

**1** Department of Ophthalmology, Pusan National University College of Medicine, Busan, South Korea, **2** Biomedical Research Institute, Pusan National University Hospital, Busan, South Korea, **3** Department of Biostatistics, Clinical Trial Center, Biomedical Research Institute, Pusan National University Hospital, Busan, South Korea

\* alertlee@naver.com

## Abstract

We developed a deep learning architecture based on Inception V3 to predict visual field using optical coherence tomography (OCT) imaging and evaluated its performance. Two OCT images, macular ganglion cell-inner plexiform layer (mGCIPL) and peripapillary retinal nerve fibre layer (pRNFL) thicknesses, were acquired and combined. A convolutional neural network architecture was constructed to predict visual field using this combined OCT image. The root mean square error (RMSE) between the actual and predicted visual fields was calculated to evaluate the performance. Globally (the entire visual field area), the RMSE for all patients was 4.79 ± 2.56 dB, with 3.27 dB and 5.27 dB for the normal and glaucoma groups, respectively. The RMSE of the macular region (4.40 dB) was higher than that of the peripheral region (4.29 dB) for all subjects. In normal subjects, the RMSE of the macular region (2.45 dB) was significantly lower than that of the peripheral region (3.11 dB), whereas in glaucoma subjects, the RMSE was higher (5.62 dB versus 5.03 dB, respectively). The deep learning method effectively predicted the visual field 24–2 using the combined OCT image. This method may help clinicians determine visual fields, particularly for patients who are unable to undergo a physical visual field exam.

## Introduction

Glaucoma is one of the leading causes of blindness in the world [1,2]. It is a widespread chronic, irreversible optic neuropathy characterised by the progressive and permanent loss of retinal ganglion cells (RGCs) and their axons. It is associated with visual field abnormalities, the loss of which can greatly impact the quality of life [3,4]. In practice, monitoring visual field examination is an important process in preventing vision loss.

However, visual field exams are very subjective tests and depend largely on patient compliance. They inherently involve several random errors and fluctuations, which can be affected by various factors, that result in a low signal-to-noise ratio [5]. The fluctuations are more severe in glaucomatous patients than in normal subjects [6,7]. It is often thought that a visual field exam is difficult to perform as several factors can affect the quality of the exam, some of which

**Data Availability Statement:** Data is provided with supporting information file. Please refer S1 File. All python source code, trained model, and test images are available on Github web site: https://github.com/climyth/VFbySD-OCT (DOI: 10.5281/

zenodo.3757702). The training images are available from the first (climyth@naver.com) or corresponding author (alertlee@naver.com) on reasonable request. The provided data must be used for only research purpose and should not be shared with other unauthorized institutions. Data will be stored in our research NAS (network attached storage) and cloud storage for backup purpose up to December 2022, after which it will be discarded as required by local regulations. Although the authors cannot make their study's data publicly available at the time of publication, all authors commit to make the data underlying the findings described in this study fully available without restriction to those who request the data, in compliance with the PLOS Data Availability policy. For data sets involving personally identifiable information or other sensitive data, data sharing is contingent on the data being handled appropriately by the data requester and in accordance with all applicable local requirements.

**Funding:** KHP, This research was supported by the Bio & Medical Technology Development Program of the National Research Foundation (NRF) funded by the South Korean government (MSIT) (No. NRF-2018M3A9E8066254).

**Competing interests:** The authors have declared that no competing interests exist.

include patient attention, fatigue, artefacts such as ptosis, lens rim defects, and incorrect refractive error correction. Despite attempts to minimise or control the influence of all these factors, a patient's learning curve may also affect the outcome of the visual field exam [8,9].

In contrast to the visual field exam, optical coherence tomography (OCT) is an objective test and its reproducibility is known to be excellent [10–15]. Patients show evidence of structural changes, including optic nerve head (ONH) damage and retinal nerve fibre layer (RNFL) thinning, before functional loss is detected by standard automated perimetry [16,17]. It is reliable when performed on both normal and glaucoma patients [18]. Structural changes measured by OCT are closely related to the functional changes in the visual field [19]. The pattern of correlation and corresponding locations between structure and function have been investigated by several previous studies [20–25]. Taken together, this suggests that it may be possible to deduce a visual field test from OCT images, which would be very helpful in monitoring patients who are unable to undergo visual field testing, including children, the elderly, and those with dementia. There have been previous attempts to predict visual fields using OCT images [26–28]; however, they used a pointwise numerical regression method.

More recently, computer technology has tremendously improved and with the aid of GPUs (graphic processing units), parallel processing capability, which is important in neural network computation, is also greatly advanced. Artificial intelligence algorithms have also improved and recently, 'deep learning algorithms' have emerged, performing at levels almost comparable to that of humans [29–31]. The biggest advantage of deep learning algorithm is that it is an end-to-end learning algorithm, i.e. a precise mechanism does not need to be provided to resolve complex problems; rather, such mechanisms are learnt during training. Structure-function relationship is a complex and non-linear problem with many unpredictable errors and large variations among patients. Neural network computation may be a good choice to deal with these types of complex problems.

The purpose of this study was to construct a deep learning architecture to predict visual fields using OCT images and evaluate its performance. We built a model using a state-of-the-art convolutional neural network (CNN) architecture and tested its accuracy globally and regionally. We also attempted to identify various factors which affected visual field prediction.

## Materials and methods

This retrospective study was performed in accordance with the tenets of the Declaration of Helsinki. The study was approved by the institutional review board (IRB) of Pusan National University Hospital, South Korea. The requirement for patient consent was waived by the IRB due to the retrospective nature of the study.

All training and test data were obtained from subjects who had visited the glaucoma clinic at Pusan National University Hospital from 2013 to 2018. The demographic characteristics of the training group are summarised in Table 1. The training dataset consisted of 2,811 eyes from 1,529 subjects and was not labelled by diagnosis. Therefore, normal fundus images, as well as data from subjects with glaucoma and other optic neuropathies, were included. However, eyes with retinal disease or severe media opacity (such as cataracts) were excluded. The mean ± standard deviation (SD) age of the test group was 62.1 ± 16.6 years. A total of 2,811 records from the training dataset was randomly split into training and validation data at a ratio of 9:1. Validation data were used to check the current fitness of the neural network during training to prevent overfitting.

In addition to the training dataset, a separate test dataset was prepared with 290 eyes from 290 subjects, with no patient overlap between datasets. For all subjects in the test group, a retrospective review of the detailed results of ophthalmic examinations was performed, including

**Table 1. Demographic characteristics of the training group.**

|  | Values |
|---|---|
| Total number of eyes | 2811 |
| Total number of patients | 1529 |
| Age (years; mean ± SD) | 62.1 ± 16.6 |
| **Number of eyes binned by visual field mean deviation (MD)** |  |
| MD > –3 dB | 1019 (36.2%) |
| –3 dB ≥ MD > –6 dB | 656 (23.3%) |
| –6 dB ≥ MD > –9 dB | 328 (11.7%) |
| –9 dB ≥ MD > –12 dB | 180 (6.4%) |
| –12 dB ≥ MD | 628 (22.3%) |

SD: standard deviation.

best corrected visual acuity (BCVA), Goldmann applanation tonometry (GAT), slit-lamp examination, funduscopy, biometry using the IOLMaster (Carl Zeiss Meditec, Dublin, CA, USA), Humphrey visual field test (Carl Zeiss Meditec), central corneal thickness (CCT) using ultrasonic pachymetry (Pachmate; DGH Technology, Exton, PA, USA), keratometry using the Auto Kerato-Refractometer (ARK-510A; NIDEK, Hiroshi, Japan), and Cirrus high definition optical coherence tomography (HD-OCT; Carl Zeiss Meditec). Glaucomatous optic neuropathy was defined if one or more of the following criteria were met: focal or diffuse neuroretinal rim thinning, localised notching, cup-to-disc ratio asymmetry ≥0.2, and the presence of retinal nerve fibre layer (RNFL) defects congruent with visual field defects [32]. Normal subjects were defined as those with no history of ocular disease, an intraocular pressure <21 mm Hg, an absence of a glaucomatous optic disc appearance, and a normal visual field. To ensure representation of the full range of disease, normal subjects also included those who were clinically suspected of having glaucoma, based on optic disc or RNFL appearance, or elevated intraocular pressure, but had normal visual field. Patients with corneal or ocular media opacity, a refractive error ≥±6.0 dioptres, optic neuropathies other than glaucoma, or recent ocular surgery or trauma were excluded.

## Spectral Domain Optical Coherence Tomography (SD-OCT)

The Cirrus spectral domain (SD)-OCT instrument (Carl Zeiss Meditec) was used to acquire macular ganglion cell-inner plexiform layer (mGCIPL) and peripapillary retinal nerve fibre layer (pRNFL) thickness maps. Two consecutive OCT exams, 6 mm × 6 mm macular cube scan 200 × 200 protocol and 6 mm x 6 mm optic disc cube 200 × 200 scan, were performed at the same time to obtain both mGCIPL and pRNFL thickness maps. Following pupil dilation using 0.5% tropicamide and 0.5% phenylephrine, the subject was seated and properly aligned. The eye was then brought into view using the mouse-driven alignment system and the line scanning image was focused by adjusting for refractive error. The macular centre or ONH was shown at the centre of the live image and, further centring (Z-offset) and enhancement were optimised. The laser scanned over a 6 mm x 6 mm square area, capturing a cube of data consisting of 200 × 200 A-scans to make B-scans (40,000 points) in about 1.5 seconds (27,000 A-scans/sec). The ganglion cell analysis algorithm automatically segmented the GCIPL and then calculated the thickness of the mGCIPL within a 6 mm x 6 mm square area centred on the fovea. The RNFL analysis algorithm automatically segmented the peripapillary RNFL layer and calculated its thickness within a 6 mm x 6 mm square area centred at the ONH centre. For quality control, only good quality scans defined as having a minimum signal strength of 6, no

involuntary eye movements, blinking artefacts, or being without misalignment or segmentation failures were used for analysis.

## Input image generation and visual field region definition

We developed a custom software to generate a combined image of mGCIPL and pRNFL thickness maps. In Fig 1, an example of the combined OCT image is shown on top of the neural network architecture. Our custom software utilised two report images exported from Cirrus OCT: 1) Ganglion Cell OU Analysis: Macular Cube 200 × 200, and 2) ONH and RNFL OU Analysis: Optic Disc Cube 200 × 200 protocol. The mGCIPL and pRNFL thickness maps from both eyes were present in the report image. The custom software automatically detected the location of these blue-toned thickness maps by searching for the rectangular boundary of the blue image, starting from a predefined location, and cropped and combined them (mGCIPL map on the left, ONH map on the right). All left eye images were flipped horizontally to match the format of the right eye.

In Fig 2A, the fundus photo with Humphrey visual field 24–2 test points and two superimposed OCT thickness maps are shown. This image was drawn by our custom software which precisely locates the visual field test points at their designated locations. The fundus photo was taken with the Nidek AFC-330 camera (Hiroshi, Japan) with a 45˚ horizontal field of view. The original fundus image had a resolution of 2438 x 2112 (width x height); however, the actual fundus area without the margin had a resolution of 2290 x 2112 (width x height), generating a linear scaling of 51 pixels per degree. Based on this information, the custom software located the visual field test points. A user manually overlapped the mGCIPL and pRNFL OCT images on the fundus photos by exactly matching retinal vessels and the shape of the ONH. In this overlapped image, we noted that the central 4 × 4 visual field test points were inside the mGCIPL scan area. We defined this central visual field area as the 'macular OCT scan area' and the surrounding area as the 'peripheral OCT scan area' (Fig 2B). We defined another set of regions using the Garway-Heath sectorisation [23]. It consisted of six sectoral areas on the ONH corresponding to the visual field test points (Fig 2B).

## Visual field examination

Within 6 months of the OCT exam, automated perimetry was performed on all training and test subjects using a Humphrey Visual Field Analyzer 750i instrument (Carl Zeiss Meditec) with the Swedish interactive threshold algorithm (SITA) 24–2 or 30–2. Of the 54 test points of the 24–2 test pattern, 2 points of physiologic scotoma were excluded and the remaining 52 test points of the total threshold value were used as the ground truth visual field of the training and test sets. The 30–2 test pattern was converted to 24–2 by using overlapping test points. Reliable visual field tests were defined as having a false-positive rate <33%, false-negative rate <33%, and fixation loss <20%. Normal subjects were defined as those with a glaucoma hemifield test (GHT) within the normal limits, and with a mean deviation (MD) and pattern standard deviation (PSD) within 95% of the normal population. Glaucomatous visual fields were those that met at least one of the following criteria: GHT outside the normal limits and/or PSD probability outside of 95% of the normal population. Glaucoma severity was determined based on the MD of the visual field test: early >–6 dB and moderate to severe ≤–6 dB.

## Deep learning architectures and training

The open source deep learning platform, Keras library [33], running on the TensorFlow™ backend (Google, Mountain View, CA, USA) python API r1.10, was used. Python language version 3.5 was used with the CUDA toolkit 9.0 and cuDNN 7.0 library to utilise the GPU

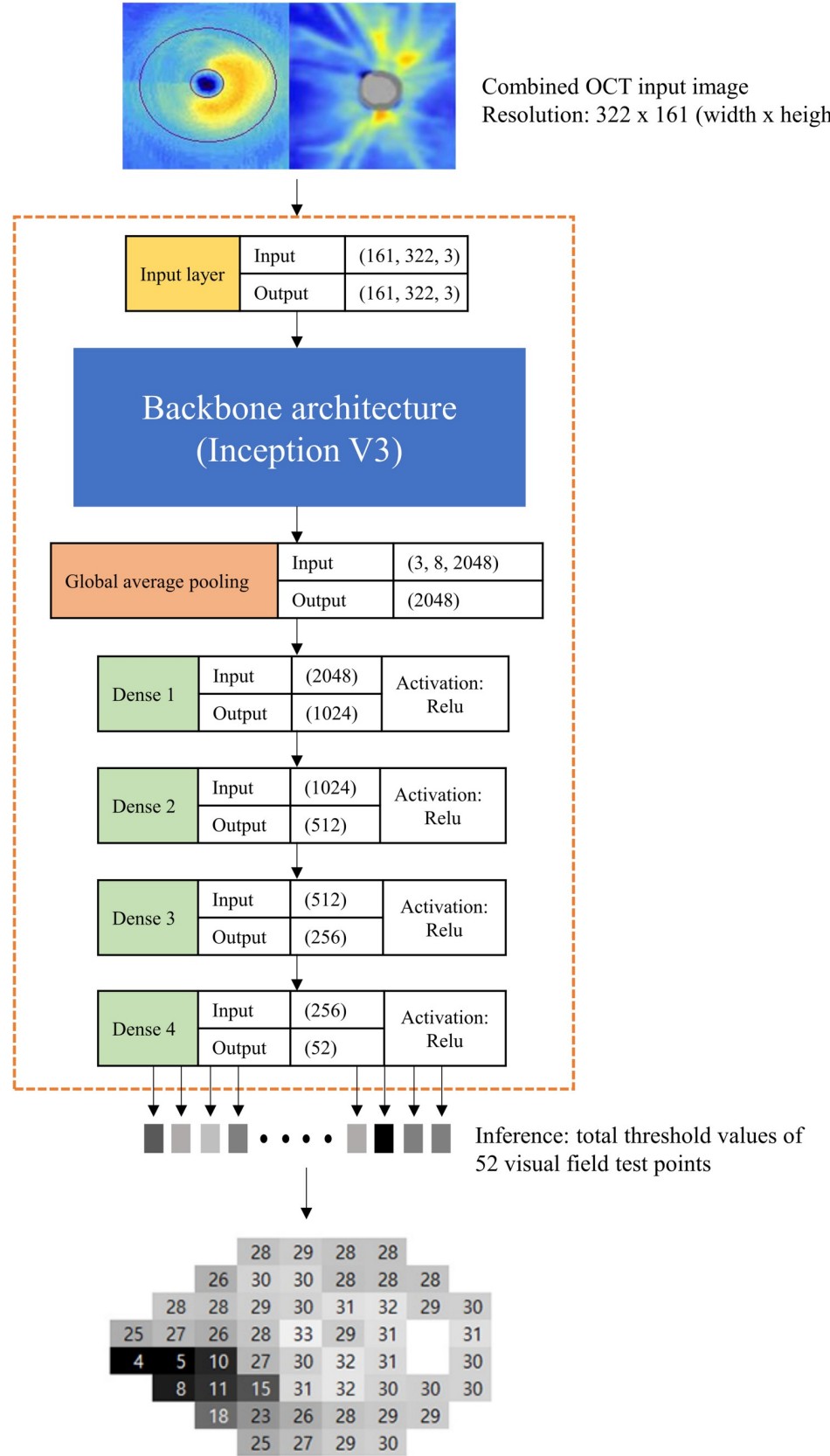

**Fig 1. Deep learning architecture.** The shape of the tensor (input/output) is described on the right side of each layer box. The global average pooling layer and four fully connected network (dense) layers were connected after the Inception V3 backbone CNN (convolutional neural network) architecture. The four dense layers used ReLu (rectified linear unit) as the activation function.

computation power. The hardware environment used for training and test runs was Intel i5-8400 CPU, 32 GB RAM, and a GeForce Titan Volta (NVIDIA, Santa Clara, CA, USA).

The final deep neural network architecture used in this study is shown in Fig 1. A state-of-the-art CNN architecture, Inception V3 [34] developed by Google, was used as the backbone structure to extract global features. A bottleneck layer of the Inception V3 was removed and replaced with one global average pooling layer followed by four consecutive densely connected

**(A)**

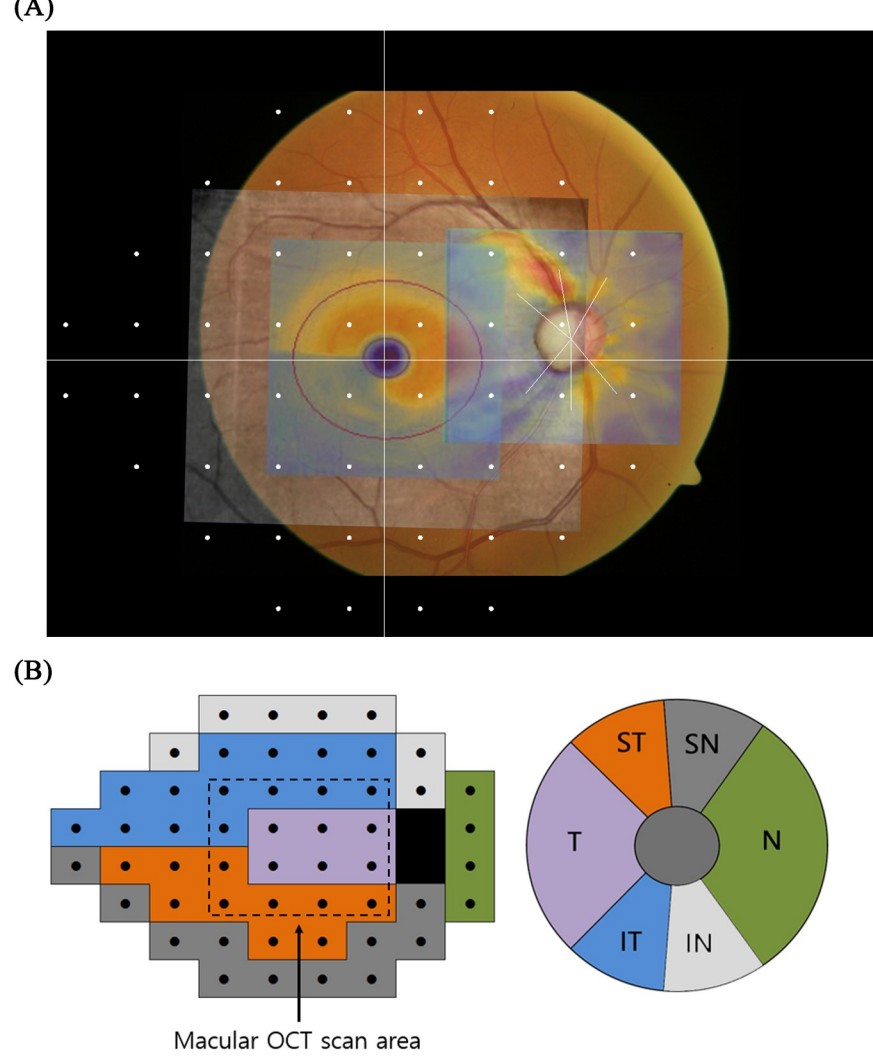

**(B)**

**Fig 2. Visual field test pattern and Garway-Heath map.** (A) A colour fundus photo with Humphrey 24–2 visual field test pattern and two optical coherence tomography (OCT) images were overlapped on the fundus photo. Garway-Heath sectorisation (white radiating line) is drawn on the optic nerve head centre. (B) Regions of visual field test points outlined by Garway-Heath sectorisation map. The central dashed square shows the boundary of the macular OCT scan area and the surrounding area is defined as the peripheral OCT scan area. IN: inferonasal, IT: inferotemporal, N: nasal, SN: superonasal, ST: superotemporal, T: temporal.

layers. All dense layers used ReLu (rectified linear unit) as the activation function. A combined OCT image, which had a size of 322 × 161 (width × height), was used as input data. The Inception V3 used the input image to produce a 3 × 8 × 2048 (height × width × depth) image containing global features. The global average pooling layer flattened the output of Inception V3 and made averaged 2048 features. Four dense layers condensed these features into 52 final output neurons which corresponded to 52 visual field threshold values (two points of physiologic scotoma were excluded from prediction).

Before training began, Inception V3 pretrained on the ImageNet dataset was downloaded and applied. No layer was frozen during training and all layers were fine-tuned. A total of 2,811 records were randomly aplit into training and validation datasets in a 9:1 ratio and batches of 64 were supplied to the neural network. The optimizer was 'rmsprop' and the loss function was 'mean squared error'. Training was monitored by reference to the loss trends of both the training and validation sets. When no further performance gain was observed over 100 epochs, training finished. To prevent overfitting, the repeated random sub-sampling cross validation technique [35] was used. The training data were again randomly split in a 9:1 ratio, the last trained weight file was loaded, and training resumed until no further performance gain was evident over 100 epochs. This process was repeated five times.

## Statistical analyses

The Shapiro-Wilk test was performed to check the normality of the data distribution. To compare parameters between normal subjects and glaucoma patients, we used Student's t-test or Mann-Whitney U test depending on the normality of the data. The chi-square test was used for categorical variables. Visual field prediction error was calculated as the root mean square error (RMSE) using the following formula:

$$RMSE = \sqrt{\sum_{n=1}^{52} \frac{(true\ THV_n - predicted\ THV_n)^2}{52}}$$

$$n = n^{th}\ test\ point\ of\ visual\ field\ exam, \quad THV = visual\ field\ threshold\ value$$

The above formula is an example of the global RMSE calculation (i.e. includes all 52 test points). When we calculated regional prediction error, only a select number of visual field test points inside the target region were used. Those regions are defined in Fig 2B.

We performed correlation analysis and simple linear regression analysis to identify factors affecting visual field prediction. Depending on data normality, Pearson's correlation coefficient or Spearman's rank correlation coefficient were used. Multiple linear regression analyses with the ENTER method were also used to identify the importance of possible factors affecting visual field prediction. For conducting statistical analyses, SPSS (version 21.0 for Windows; SPSS, Chicago, IL, USA) and MedCalc (version 12.5 for Windows; Ostend, Belgium) were used, and $P < 0.05$ (single comparison) and $P < 0.017$ (multiple comparisons) were considered to indicate statistical significance.

## Results

A total of 290 eyes from 290 subjects were recruited for the test group, including 112 normal subjects (60 normal, 52 suspected glaucoma), and 178 glaucoma patients (115 early stage, 63 moderate to severe glaucoma). The demographic characteristics are summarised in Table 2. Although the visual acuity (logarithm of the minimum angle of resolution; logMAR) and spherical equivalence were significantly different ($P = 0.027$, $P = 0.006$ respectively) between

**Table 2. Demographic characteristics of the test group.**

|  | Normal (n = 112) | Glaucoma (n = 178) | P value |
|---|---|---|---|
| Age (years) | 52.5 ± 15.4 | 51.7 ± 14.0 | 0.537[a] |
| Female / male (number) | 56 / 56 | 85 / 93 | 0.780[b] |
| Visual acuity (logMAR) | 0.066 ± 0.112 | 0.093 ± 0.121 | 0.027[a] |
| Spherical equivalence (dioptre) | −1.44 ± 2.84 | −2.36 ± 3.13 | 0.006[a] |
| Intraocular pressure (mm Hg) | 15.6 ± 3.9 | 15.6 ± 4.1 | 0.744[a] |
| Axial length (mm) | 24.34 ± 1.62 | 24.70 ± 1.65 | 0.058[a] |
| Central corneal thickness (μm) | 548.8 ± 37.8 | 548.9 ± 32.6 | 0.917[a] |
| **Visual field test** |  |  |  |
| • Mean deviation (dB) | −1.62 ± 2.10 | −6.29 ± 6.18 | < 0.001[a] |
| • Pattern standard deviation (dB) | 1.98 ± 1.11 | 6.12 ± 4.10 | < 0.001[a] |
| • Visual field index (%) | 97.8 ± 3.2 | 84.2 ± 18.8 | < 0.001[a] |
| **Optical coherence tomography** |  |  |  |
| • Average mGCIPL thickness (μm) | 79.3 ± 6.9 | 69.2 ± 9.0 | < 0.001[a] |
| • Average pRNFL thickness (μm) | 91.0 ± 10.2 | 72.5 ± 12.4 | < 0.001[a] |

Values are presented as mean ± standard deviation.

mGCIPL: macular ganglion cell-internal plexiform layer, pRNFL: peripapillary retinal nerve fibre layer.

[a] Mann-Whitney U test.

[b] χ2 test.

the normal subjects and glaucoma patients, age, gender, intraocular pressure, axial length, and CCT were not. All visual field parameters were also significantly different between these two groups, including average visual field MD which was -1.62 dB for normal subjects and -6.29 dB for glaucoma patients. The average OCT parameters were also significantly different with average mGCIPL thickness measuring 79.3 μm and 69.2 μm, and average pRNFL thickness measuring 91.0 μm and 72.5 μm, respectively.

Global and regional visual field prediction error and representative examples of prediction are shown in Table 3 and Fig 3. Globally (the entire visual field area), the RMSE was 4.79 ± 2.56 dB including all patients and, 3.27 dB and 5.27 dB in normal subjects and glaucoma patients, respectively. The prediction error in normal subjects was always significantly lower than that in glaucoma patients (all $P < 0.001$), regardless of regions. By sector, the prediction error of the superior visual field region was generally lower than that of the corresponding inferior visual field region. The nasal region showed the lowest prediction error (3.50 dB) followed by the superotemporal (3.93 dB), temporal (4.08 dB), superonasal (4.55 dB), inferotemporal (4.66 dB), and inferonasal (5.23 dB) regions. In comparing the OCT scan areas of all subjects, the prediction error of the macular region (4.40 dB) was significantly higher than that of the peripheral region (4.29 dB; $P = 0.031$, Mann-Whitney U test). However, in normal subjects, the prediction error of the macular region (2.45 dB) was significantly lower ($P < 0.001$, Mann-Whitney U test) than that of the peripheral region (3.11 dB), whereas in glaucomatous patients, the prediction error of macular region (5.62 dB) was higher than that of the peripheral region (5.03 dB), though not significantly ($P = 0.741$, Mann-Whitney U test).

The representative example of the class activation map (CAM) is shown in Fig 4. In this map, the red colour denoted the area where the CNN was highly activated and produced a high sensitivity value for the visual field test point, whereas the blue colour denoted the opposite. The actual visual field test result (Fig 4A) showed low sensitivity in the superonasal area and the CAM images at the corresponding location in the collection (Fig 4C) showed low activation in the inferotemporal sectors of the ONH OCT scan image. In contrast, the inferonasal

Table 3. Global and regional root mean square error of visual field prediction.

| | All subjects | Subject group | | |
| | | Normal | Glaucoma | P value[a] |
|---|---|---|---|---|
| Global | 4.79 ± 2.56 | 3.27 ± 1.50 | 5.75 ± 2.63 | <0.001 |
| **Region by Garway-Heath sectorisation** | | | | |
| Superotemporal | 3.93 ± 3.53 | 2.41 ± 1.63 | 4.88 ± 4.04 | <0.001 |
| Temporal | 4.08 ± 3.34 | 2.36 ± 1.36 | 5.16 ± 3.74 | <0.001 |
| Inferotemporal | 4.66 ± 3.14 | 3.07 ± 1.53 | 5.66 ± 3.46 | <0.001 |
| Superonasal | 4.55 ± 2.92 | 3.30 ± 1.93 | 5.33 ± 3.16 | <0.001 |
| Nasal | 3.50 ± 2.56 | 2.69 ± 1.72 | 4.01 ± 2.86 | <0.001 |
| Inferonasal | 5.23 ± 2.92 | 4.41 ±2.39 | 5.74 ± 3.10 | <0.001 |
| **Region by OCT scan area** | | | | |
| Macular[b] | 4.40 ± 3.26 | 2.45 ± 1.20 | 5.62 ± 3.54 | <0.001 |
| Peripheral[c] | 4.29 ± 2.20 | 3.11 ± 1.47 | 5.03 ± 2.27 | <0.001 |

Values are presented as mean ± standard deviation.

OCT: optical coherence tomography.

[a] Mann-Whitney U test between normal and glaucoma group.

[b] Visual field test points inside the macular OCT scan area.

[c] Visual field test points outside the macular OCT scan area.

area showed high values for visual field sensitivity and the CAM images at the corresponding location showed high activation (seen as the red colour) in the superotemporal sectors of the ONH OCT scan image.

In each CAM image, red indicates strongly activated points yielding high threshold values; blue (or no color) indicates the opposite. In this example, the visual field damage is principally superonasal; the inferonasal region is relatively intact (i.e., exhibits high threshold values). The CAM images in (C) that are numbered 27~30, 35~38, 43~46, and 49~52 are intensely red (high visual field threshold values). Note that these activated areas in the CAM images (also intensely red) exactly match the orange and dark grey regions in (D). In contrast, the CAM images numbered 5, 6, 11, 12, and 19~22 are not colored (and thus not activated) and generate low threshold visual field values. These areas match the blue and light grey regions in (D).

OCT: optical coherence tomography, ONH: optic nerve head.

Correlation analysis (Spearman's rho) was performed to determine factors affecting visual field prediction (Table 4). Age, sex, spherical equivalence, CCT, axial length, and macular OCT signal strength all showed no significant correlation with visual field prediction. Visual acuity (logMAR) positively correlated (r = 0.157, P = 0.007), and visual field MD (r = –0.543, P < 0.001), average mGCIPL thickness (r = –0.553, P < 0.001), ONH OCT signal strength (r = –0.126, P = 0.032), and average pRNFL thickness (r = –0.597, P < 0.001) negatively correlated with visual field prediction.

Multiple linear regression analysis was performed to investigate the relative influence of possible factors affecting visual field prediction (Table 5). The model was constructed using the ENTER method and the RMSE as the outcome variable. Age, visual acuity (logMAR), spherical equivalence, CCT, axial length, visual field MD, macular OCT signal strength, average mGCIPL thickness, ONH OCT signal strength, and average pRNFL thickness were used as the input variables. The final model had $R^2$ = 0.463 and P < 0.001. No multicollinearity was found between variables (all variance inflation factors, VIFs ≤ 3.051). Three out of 10 input variables were significantly correlated with prediction error. The visual field MD was the most

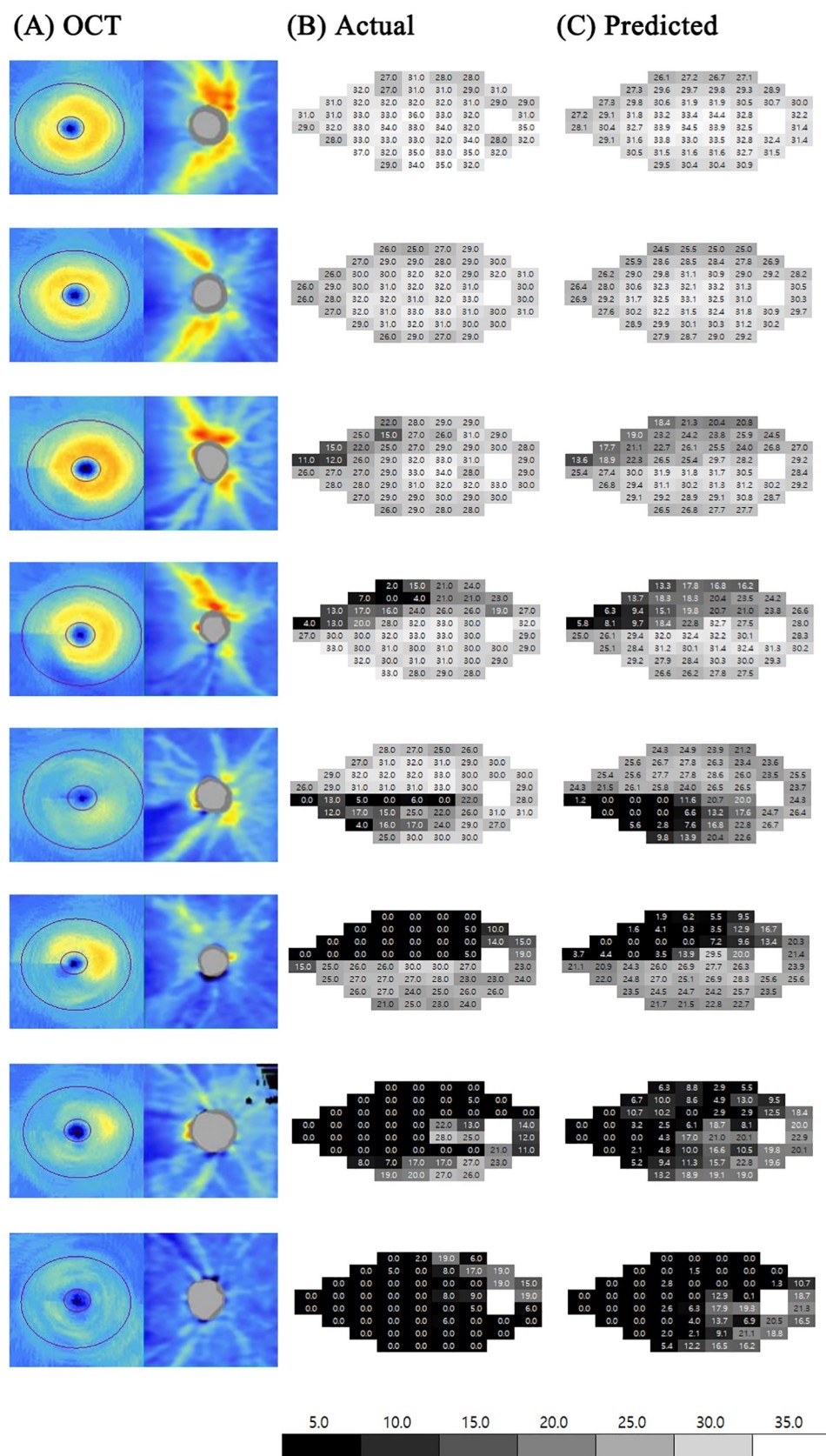

**Fig 3. Representative cases of visual field prediction.** (A) The combined OCT images, which were input into the deep learning architecture, are shown on the left column. The actual threshold values of visual field exams are shown in the (B) middle panel and the threshold values predicted by Inception V3 based deep learning architecture are shown on the (C) right panel. The colour reference for the threshold values are shown at the bottom. Despite the artificial intelligence having never seen the actual visual field, the predicted visual field looked very similar to the actual visual field exam.

influential variable (β = –0.433, $P < 0.001$) followed by average pRNFL thickness (β = –0.252, $P = 0.002$) and average mGCIPL thickness (β = –0.170, $P = 0.028$).

Fig 5 shows the relationship between prediction error ratio and the visual field MD using a scatter plot. The prediction error was defined as the ratio of average prediction error inside the macular OCT scan area (Fig 2) divided by the prediction error in the peripheral scan area (i.e. outside the macular OCT scan area). In linear regression analysis, the prediction error ratio was negatively associated with the visual field MD and its slope was –0.020 ($P < 0.001$). In other words, as the MD decreased, the macular prediction error became greater than the peripheral prediction error.

## Discussion

The main objective of this study was to develop a deep learning architecture to predict Humphrey visual field 24–2 threshold values from macular and ONH OCT imaging. We evaluated the performance of visual field prediction globally and regionally, and tried to identify the factors that affected prediction. Globally, the RMSE of the deep learning algorithm was 4.70 ± 2.56 dB for all test subjects. In glaucoma subjects, the prediction error was significantly higher than that of normal subjects (5.75 dB versus 3.27 dB, respectively). Visual field MD was the most influential factor for prediction, followed by average pRNFL and mGCIPL thickness. In the ophthalmology field, this study is the first to use OCT and a deep learning algorithm to predict Humphrey visual field 24–2.

Retinal ganglion cells and their nerve fibre layer are closely related to the glaucomatous visual field defect. This structure-function relationship has been investigated extensively. Wollstein et al. [20] studied the relationship between OCT-measured macular retinal thickness, pRNFL thickness, and visual field. In their report, macular retinal thickness was able to detect glaucomatous visual field damage and also correlated with pRNFL thickness. Sato et al. [21] reported that GCIPL thickness, measured by Cirrus HD-OCT, was significantly correlated with the central visual field. Similarly, Raza et al. [22] showed that GCIPL thickness was well correlated with visual field loss within 7.2˚ of the fovea. Kim et al. [36] noted that both macular ganglion cell complex (GCC) thickness and pRNFL thickness showed similar diagnostic performance in detecting glaucoma. In our previous study [25], we found that the macular OCT scan area mostly overlapped with visual field 10–2 test points and closely correlated with each other. Wu et al. [37] reported that localised pRNFL thinning measured by SD-OCT was well correlated with localised glaucomatous visual field defects. Garway-Heath et al. [23] mapped visual field locations to the ONH sectors derived by overlapping visual field test points on RNFL photographs. Gardiner et al. [24] also showed a topographical map between visual field locations and ONH sectors. In those previous reports, mGCIPL thickness was more related to the central visual field whereas pRNFL thickness was related to gross glaucomatous change. Therefore, we could theorise that these two OCT exams may have complementary roles in predicting visual field.

Although attempts to use machine learning algorithms for structure-function relationships in glaucoma are not novel [38–40], there have been few studies that predict visual field using OCT imaging. In a recent study similar to ours, Christopher et al. [41] used a deep learning

**(A) Actual**

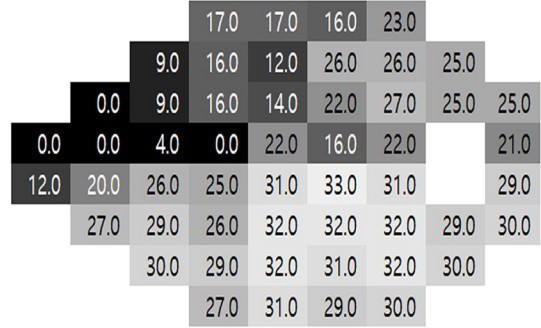

**(B) Predicted**

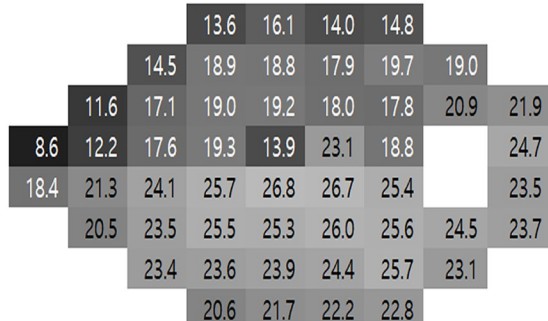

**(C) Collection of class activation maps**

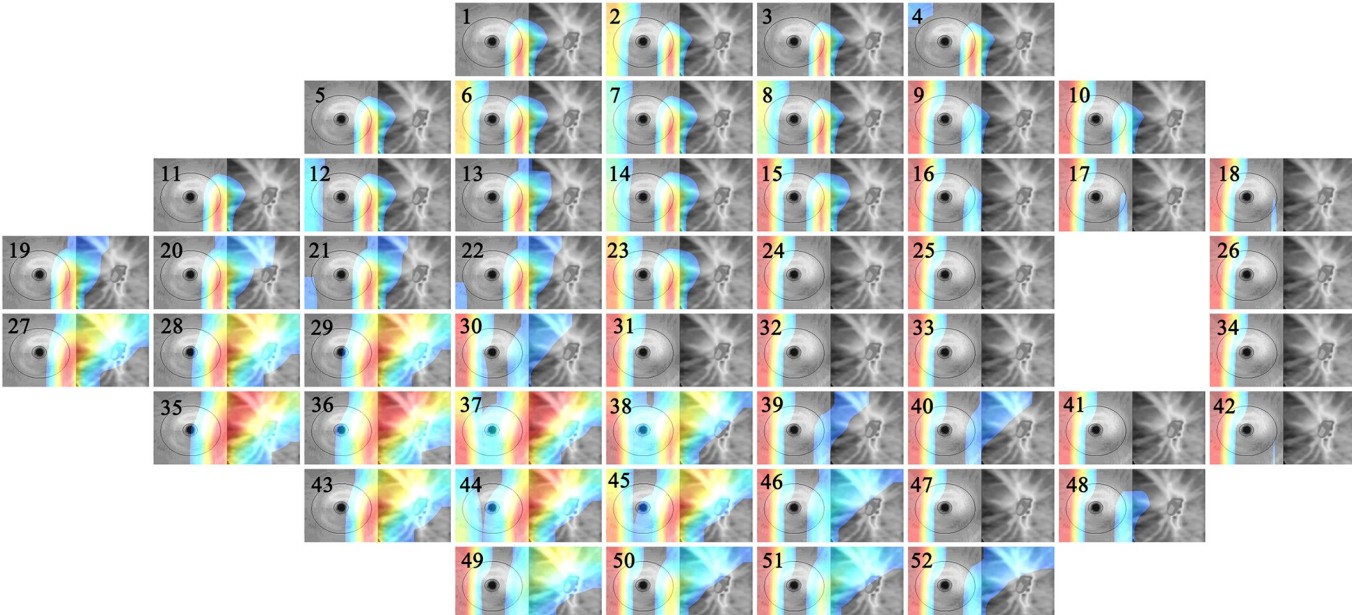

**(D) Structure-function mapping between combined OCT image and visual field**

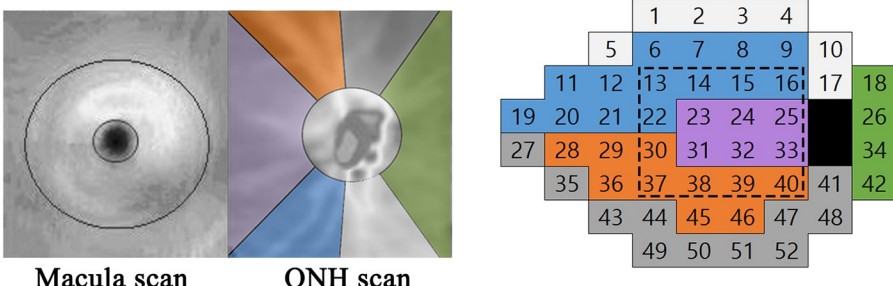

Macula scan          ONH scan

**Fig 4. Representative example of Class Activation Map (CAM).** The figure shows (A) the actual threshold values and (B) the predicted threshold values of the visual field examination. (C) Fifty-two CAMs were placed at individual visual field test points. Each CAM image is numbered at the top left. (D) Structure-function mapping between the combined OCT image (left) and the visual field (right). The macular scan in the combined OCT image corresponds to the dashed rectangle in the visual field. Color-coded Garway-Heath sectors are superimposed on the ONH scan of the combined OCT image and the corresponding visual field regions are similarly colored. The numbers in the visual field image are the same as those in the CAM images of (C).

**Table 4. Correlation coefficients and simple linear regression analyses between visual field prediction error and various factors.**

| | Correlation coefficients | | Simple linear regression analysis | | | |
|---|---|---|---|---|---|---|
| | Spearman's rho | *P* value | Slope | Intercept | $R^2$ | *P* value |
| Age | −0.003 | 0.957 | 0.011 | 4.230 | 0.004 | 0.301 |
| Sex | −0.018 | 0.756 | −0.108 | 4.842 | <0.001 | 0.722 |
| Visual acuity (logMAR) | 0.157 | 0.007 | 2.803 | 4.556 | 0.017 | 0.028 |
| Spherical equivalence | −0.018 | 0.765 | 0.015 | 4.818 | <0.001 | 0.766 |
| Central corneal thickness | −0.051 | 0.404 | −0.003 | 6.705 | 0.002 | 0.452 |
| Axial length | 0.071 | 0.257 | 0.015 | 4.399 | <0.001 | 0.878 |
| Visual field MD | −0.543 | <0.001 | −0.296 | 3.463 | 0.403 | <0.001 |
| Macular OCT signal strength | −0.039 | 0.510 | −0.103 | 5.625 | 0.002 | 0.410 |
| Average mGCIPL thickness | −0.553 | <0.001 | −0.145 | 15.415 | 0.296 | <0.001 |
| ONH OCT signal strength | −0.126 | 0.032 | −0.303 | 7.280 | 0.019 | 0.020 |
| Average pRNFL thickness | −0.597 | <0.001 | −0.102 | 12.896 | 0.338 | <0.001 |

MD: mean deviation, mGCIPL: macular ganglion cell-internal plexiform layer, OCT: optical coherence tomography, ONH: optic nerve head, pRNFL: peripapillary retinal nerve fibre layer.

method to predict glaucomatous visual fields from OCT images. The deep learning architecture employed to predict visual field global indices was ResNet and the inputs were Spectralis SD-OCT ONH images. Various image types (RNFL thickness maps, RNFL en-face images, and confocal scanning laser ophthalmoscopic images) were input and the predictions using each type were compared. Unlike our deep learning method, which predicts the entire visual field from both macular and ONH images, the method of Christopher et al. uses only ONH image as input and predicts visual field global indices including the mean deviation (MD), the pattern standard deviation (PSD), and the mean sectoral pattern deviation. The best mean absolute errors, between the real and predicted values, were 2.5 dB (MD) and 1.5 dB (PSD). Zhu et al. [42] introduced a method akin to the neural network with a radial basis function customised under a Bayesian framework (BRBF), to predict visual field from pRNFL thickness. In their report, the mean absolute error of the BRBF was 2.9 dB, which was better than the

**Table 5. Multiple linear regression analyses between visual field prediction error and various factors.**

| | Adjusted β | *P* value | VIF |
|---|---|---|---|
| Age | −0.081 | 0.216 | 1.983 |
| Visual acuity (logMAR) | −0.059 | 0.253 | 1.204 |
| Spherical equivalence | 0.130 | 0.081 | 2.558 |
| Central corneal thickness | −0.007 | 0.889 | 1.065 |
| Axial length | 0.020 | 0.772 | 2.159 |
| Visual field MD | −0.433 | 0.000 | 1.846 |
| Macular OCT signal strength | 0.097 | 0.127 | 1.849 |
| Average mGCIPL thickness | −0.170 | 0.028 | 2.731 |
| ONH OCT signal strength | 0.040 | 0.540 | 1.914 |
| Average pRNFL thickness | −0.252 | 0.002 | 3.051 |

Final model: Outcome = root mean square error (RMSE) of prediction. Adjusted $R^2$ = 0.463, $P < 0.001$, ENTER method used. No multicollinearity was found between variables (all VIFs ≤ 3.051).

MD: mean deviation, mGCIPL: macular ganglion cell-internal plexiform layer, OCT: optical coherence tomography, ONH: optic nerve head, pRNFL: peripapillary retinal nerve fibre layer, VIF: variance inflation factors.

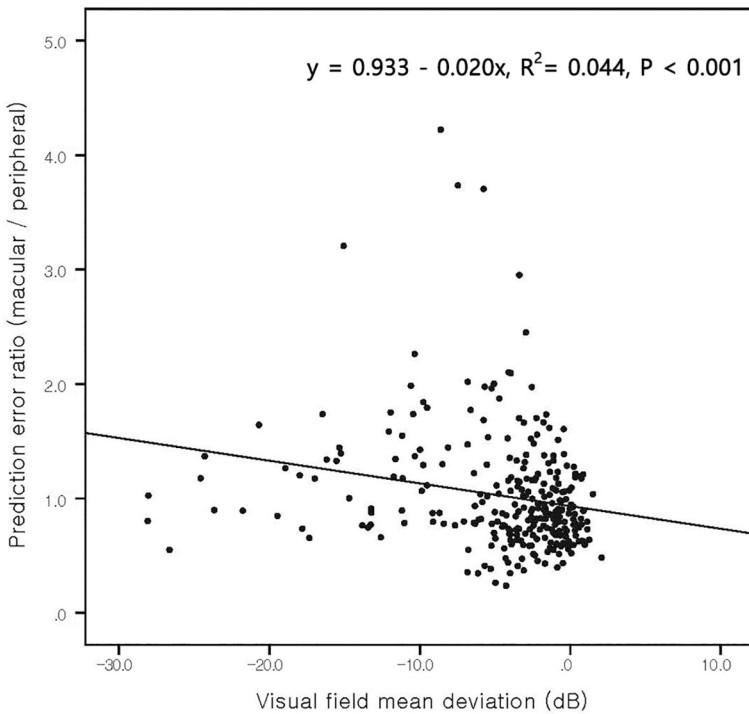

**Fig 5. Scatter plot of the prediction error ratio (macular/peripheral) versus the visual field Mean Deviation (MD).**
The slope was –0.020 (P < 0.001) which suggested that as the MD decreased, the macular prediction error became higher than the peripheral prediction error. In other words, as glaucoma progressed, the peripheral prediction became more accurate than the macular prediction.

classical linear regression model (4.9 dB). Though their result showed better performance than that noted in our study (4.70 dB), absolute comparison was not possible as the performance metric used in their study was mean absolute error and their test dataset (Blue Mountains Eye Study [BMES] data) largely consisted of healthy subjects (230 healthy subjects and 76 glaucoma patients). Considering that the prediction error was worse in glaucoma patients than normal subjects, the large proportion of healthy subjects in their study likely reduced their prediction error compared to that determined in this study.

Another study using machine learning to predict visual field threshold values was conducted by Guo et al. [26], who predicted visual field 24–2 from wide field composite OCT. They used a 9-field per eye protocol which fixated the patient's eye on a 3 × 3 grid spot pattern to obtain a total of nine OCT images. These images were stitched together to generate a single wide field composite OCT image. They constructed four predefined topological structure-function maps and applied a support vector machine (SVM) algorithm to these maps to predict visual field. Among the four predefined maps, the best map showed an RMSE of 5.42 dB. The performance of this method was excellent, but depended largely on how the structure-function map was defined. In the different maps, the prediction error increased up to 7.24 dB. However, our deep learning method creates a structure-function map by itself during the training process. In Fig 4C, the CAM showed how Inception V3 based deep learning architecture constructed this map and noted that it was similar to previous studies such as the Garway-Heath map. Moreover, the deep learning algorithm not only considers a specific mapping spot but also broad neighbouring areas as well. This likely made the prediction more accurate as a predefined mapping spot could contain errors; however, by considering a wider area, this error can be overcomed.

Among the regions of the Garway-Heath sectorisation, we observed that the superior sectors of the ONH had fewer prediction errors than the corresponding inferior sectors, suggesting that the superior retina is better correlated with functional tests than the inferior retina. This is corroborated by similar findings reported by Guo et al. [26], who found that the correlation between structure and function was higher in the superior than the inferior retina. It was suggested that this was due to the superior retina, which is responsible for the inferior visual field, being more important for survival in nature and may be an evolutionary consequence. However, we propose another possible reason for this observation. From previous studies, glaucomatous damage is known to occur sequentially in sectors. It begins in the inferotemporal ONH region, and then progresses to the superotemporal sectors. [43] In our study, as glaucoma progressed, the overall prediction error was increased. Since the inferotemporal ONH region is the first to be damaged, its prediction error could be higher than that of the superior region.

In regression analysis, we found that both mGCIPL and pRNFL thickness are significantly correlated to the visual field prediction error with the pRNFL thickness being slightly more influential than mGCIPL thickness (adjusted β = –0.170 and –0.252 for mGCIPL and pRNFL, respectively). We suggest that this is because the mGCIPL only provides information regarding the macular area whereas the pRNFL offers a more generalised view across all areas of the retina. However, the regional prediction errors, and macular and peripheral OCT scan areas were different between normal and glaucoma subjects. In normal subjects, the macular OCT scan area showed a lower prediction error than the peripheral OCT scan area, whereas in glaucoma patients, the opposite was noted. This result is consistent with previous studies. Wollstein et al. [20] reported that the pRNFL thickness was more sensitive to glaucomatous damage. Kim et al. [36] reported that mean GCC thickness, instead of pRNFL thickness, was a better diagnostic indicator of early glaucoma cases as well as a non-glaucomatous condition. In normal or early glaucoma patients, information provided by the mGCIPL is relatively more important than that provided by the pRNFL which is likely to have made visual field prediction using the macular OCT scan area more accurate. As glaucoma progressed, the information provided by the pRNFL thickness became more influential and reversed the prediction accuracy between the macular and peripheral OCT scan areas. This may provide an explanation for the negative correlation of the prediction error ratio (macular versus peripheral) with visual field MD in Fig 5.

Regression analysis also revealed that age, visual acuity, spherical equivalence, axial length, and signal strength of OCT were not correlated with prediction error. In several previous reports, the reproducibility of OCT imaging has been considered to be generally reliable [10–15]. Axial length has been shown to affect the RNFL thickness measurement [44]; however, another study has noted that both macular GCC and pRNFL thickness measurements show good diagnostic performance in individuals with high myopia [45]. In our study, extreme cases including spherical equivalence >6.0 dioptres and axial length >26.0 mm were excluded, and only eyes with relatively good visual acuity, no media opacity, and no diseases other than glaucoma were included. With these inclusion criteria, the reproducibility of OCT imaging in predicting visual field appears to be strong and negligibly affected by the factors noted above. Even though the quality of the OCT scan is inevitably associated with the accuracy of visual field prediction, there was no significant correlation between OCT image quality and prediction error.

This study had a few limitations. First, only those patients with glaucoma were included to predict visual field. If patients with diseases other than glaucoma, where the visual field defect was not altitudinal in nature, such as temporal hemianopsia, were included, the prediction accuracy may be different than that reported in our study. Second, the training and test

datasets were comprised of a primarily Korean population and the possibility of performance being different in populations comprising of other ethnicities should be considered. Girkin et al. [46] evaluated the ONH, RNFL, and macular parameters yielded by SD-OCT in terms of age and race; all parameters varied by race. Thus, researchers must be aware that visual field predictions derived from SD-OCT will also vary by race.

In conclusion, the deep learning method effectively predicted the visual field 24–2 using the combined OCT image (mGCIPL and pRNFL thickness map) with a prediction error of 4.70 dB. The accuracy of the visual field prediction was not influenced by factors such as age, visual acuity, spherical equivalence, axial length, and OCT signal strength. This may help clinicians perform visual field testing, especially of patients who are unable to undergo real visual field examinations (young children, dementia patients, and mentally retarded patients).

## Supporting information

**S1 File. All test dataset.**
(XLSX)

## Author Contributions

**Conceptualization:** Keunheung Park.

**Data curation:** Keunheung Park, Jinmi Kim.

**Formal analysis:** Keunheung Park.

**Funding acquisition:** Keunheung Park.

**Investigation:** Keunheung Park.

**Methodology:** Keunheung Park, Jinmi Kim.

**Project administration:** Jiwoong Lee.

**Resources:** Keunheung Park.

**Software:** Keunheung Park.

**Supervision:** Jiwoong Lee.

**Validation:** Keunheung Park.

**Visualization:** Keunheung Park.

**Writing – original draft:** Keunheung Park.

**Writing – review & editing:** Jiwoong Lee.

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
