## [Decision Letter · Decision Letter 0]

11 Mar 2020

PONE-D-19-26236

A deep learning approach to predict visual field using optical coherence tomography

PLOS ONE

Dear Dr. Lee,

Thank you for submitting your manuscript to PLOS ONE. After careful consideration, we feel that it has merit but does not fully meet PLOS ONE’s publication criteria as it currently stands. Therefore, we invite you to submit a revised version of the manuscript that addresses the points raised during the review process.

We would appreciate receiving your revised manuscript by Apr 25 2020 11:59PM. To enhance the reproducibility of your results, we recommend that if applicable you deposit your laboratory protocols in protocols.io, where a protocol can be assigned its own identifier (DOI) such that it can be cited independently in the future. For instructions see: http://journals.plos.org/plosone/s/submission-guidelines#loc-laboratory-protocols

We look forward to receiving your revised manuscript.

Kind regards,

Ireneusz Grulkowski, PhD

Academic Editor

PLOS ONE

Journal Requirements:

2) We note that you have indicated that data from this study are available upon request. PLOS only allows data to be available upon request if there are legal or ethical restrictions on sharing data publicly. For information on unacceptable data access restrictions, please see http://journals.plos.org/plosone/s/data-availability#loc-unacceptable-data-access-restrictions.

Reviewers' comments:

Reviewer's Responses to Questions

**Comments to the Author**

1. Is the manuscript technically sound, and do the data support the conclusions?

Reviewer #1: Yes

Reviewer #2: Yes

2. Has the statistical analysis been performed appropriately and rigorously? 

Reviewer #1: Yes

Reviewer #2: Yes

3. Have the authors made all data underlying the findings in their manuscript fully available?

Reviewer #1: No

Reviewer #2: No

4. Is the manuscript presented in an intelligible fashion and written in standard English?

Reviewer #1: Yes

Reviewer #2: Yes

5. Review Comments to the Author

Reviewer #1: The paper is well-written, easy-to-understand and ordered reasonably. The statistical methods are correct and reliable and deep learning method is easy but accurate enough.

My main criticism is about literature review which lacks in deep learning section, but mostly explores different methods for estimating VF using OCT information. A very similar paper which should be explored thoroughly is: Deep Learning Approaches Predict Glaucomatous Visual Field Damage from OCT Optic Nerve Head En Face Images and Retinal Nerve Fiber Layer Thickness Maps by Mark Christopher. The authors should provide explanation on similarities and differences of their method compared to this work. They should also compare the results with their outcomes. It is also essential to site more papers about similar applications using deep learning to show the variety of possible application with deep learning, among which one may consider estimation of VF values.

The second issue is regarding the proposed deep learning structure which needs some more explanation. Loss functions and activation maps in each layer, number of frozen layers in fine tuning stage, and comparison with other possible network structures should be elaborated in more detail. Furthermore, depicting some plots from learning curves may be informative to the reader.

Finally, the authors should provide more detailed explanation on production of class activation maps; namely, the information from maps in each regression output should be explained and compared with simple classification approaches.

Reviewer #2: The authors present a deep learning approach in order to predict visual fields from optical coherence tomography (OCT) imaging. Based on the Google Inception V3 architecture, the authors trained their network on 1529 subjects (2811 eyes) and tested it on 290 subjects (290 eyes, 112 normal subjects and 178 glaucoma patients). Performance was evaluated using the RMSE, showing significant differences between normal subjects and patients with glaucoma (3.27 to 5.75). Importantly, the performance was also evaluated for different regions (Garway-Heath sectorisation and OCT scan area), in order to assess the regional specificity of the analysis. In addition, the authors perform many additional analyses with respect to the influence of different covariates including age, visual acuity, spherical equivalence etc. The study seem to be technically sound and conclusions are drawn appropriately based on the data presented. The discussion is comprehensive and covers related research. However, I would like to make the following comments:

Comments:

- What is the clinical significance of this study? Why should a deep learning network be used instead of an automated perimetry? In lines 414/415, the authors write that such a system could be useful for patients that are "unable to undergo an actual visual field exam". What would be the potential settings?

- Some methodological details are missing on the way of fine-tuning and training process (lines 191-197): How did the authors perform the fine-tuning? And why is this opposed to transfer learning? Transfer learning can also be used in the framework of fine-tuning (e.g. weights of the pretrained network can be used as initialization for the new data, weights can be fixed in different layers). What kind of heuristic did the authors use for avoiding overfitting? Could the authors be a bit more specific about "When no more accuracy gains were observed..." (over a certain window size? How do the training and valdiation curve look like? How much variability?)?

- No availability of code: Authors should make their code publicly available (e.g. via github) so that others can test the code on their data.

- Availability of data: Authors state that the data is available on reasonable request. What do the authors mean by that? Are there any reasons (e.g. privacy, no consent) speaking against publishing the data?

- The authors write in the abstract that they "developed a novel deep learning architecture", but the architecture is a variation of an existing architecture (Inception V3).

- Why did the authors not evaluate the influence of sex in Table 4?

- Why did the authors use class activation maps in order to visualize important features? What about other explainability techniques (e.g. sensitivity analyses, occlusion or layer-wise relevance propgation)?

- I would recommend not using the term "artificial intelligence" (e.g., line 251, lines 363/364), but be more specific.

- Line 66: Cite a few papers showing that deep learning performed comparable to humans.

- Line 66: Machine learning? Especially deep learning allows for end-to-end learning. In classical machine learning, features are extracted beforehand.

- Line 92: Test dataset consists of 290 eyes of 290 subjects? Why only one eye per subject?

- Lines 409/410: Would the authors expect that results are not transferable to people from other ethnicities (just out of curiosity)? But it is a good point to make here!

6. PLOS authors have the option to publish the peer review history of their article (what does this mean?). If published, this will include your full peer review and any attached files.

Reviewer #1: Yes: Dr Rahele Kafieh

Reviewer #2: Yes: Kerstin Ritter

---

## [Author Response · Author response to Decision Letter 0]

20 Apr 2020

Dear editorial office. 

Thank you for your careful consideration of our manuscript. You sent an e-mail requesting DOI number for our Github repository. Per your request, we created DOI: 10.5281/zenodo.3757702 and modified our manuscript to address this DOI number.

---

## [Decision Letter · Decision Letter 1]

5 Jun 2020

A deep learning approach to predict visual field using optical coherence tomography

PONE-D-19-26236R1

Dear Dr. Lee,

We’re pleased to inform you that your manuscript has been judged scientifically suitable for publication and will be formally accepted for publication once it meets all outstanding technical requirements.

Kind regards,

Ireneusz Grulkowski, PhD

Academic Editor

PLOS ONE

Additional Editor Comments (optional):

Reviewers' comments:

Reviewer's Responses to Questions

**Comments to the Author**

1. If the authors have adequately addressed your comments raised in a previous round of review and you feel that this manuscript is now acceptable for publication, you may indicate that here to bypass the “Comments to the Author” section, enter your conflict of interest statement in the “Confidential to Editor” section, and submit your "Accept" recommendation.

Reviewer #1: All comments have been addressed

2. Is the manuscript technically sound, and do the data support the conclusions?

Reviewer #1: Yes

3. Has the statistical analysis been performed appropriately and rigorously? 

Reviewer #1: N/A

4. Have the authors made all data underlying the findings in their manuscript fully available?

Reviewer #1: Yes

5. Is the manuscript presented in an intelligible fashion and written in standard English?

Reviewer #1: Yes

6. Review Comments to the Author

Reviewer #1: All comments are answered perfectly. The authors made a lot of change in their figure, presentation and more importantly they argued similarities and differences with a specific paper that I introduced.The paper can be accepted now.

7. PLOS authors have the option to publish the peer review history of their article (what does this mean?). If published, this will include your full peer review and any attached files.

Reviewer #1: Yes: Rahele Kafieh

---

## [Editor Report · Acceptance letter]

22 Jun 2020

PONE-D-19-26236R1 

A deep learning approach to predict visual field using optical coherence tomography 

Dear Dr. Lee:

I'm pleased to inform you that your manuscript has been deemed suitable for publication in PLOS ONE. Congratulations! Your manuscript is now with our production department. 

Kind regards, 

on behalf of

Dr. Ireneusz Grulkowski 

Academic Editor

PLOS ONE